# Aqueous Extract from *Cuminum cyminum* L. Seed Alleviates Ovalbumin-Induced Allergic Rhinitis in Mouse via Balancing of Helper T Cells

**DOI:** 10.3390/foods11203224

**Published:** 2022-10-15

**Authors:** Momoko Ishida, Fuka Miyagawa, Kosuke Nishi, Takuya Sugahara

**Affiliations:** 1Department of Bioscience, Graduate School of Agriculture, Ehime University, Matsuyama 790-8566, Ehime, Japan; 2Food and Health Sciences Research Center, Ehime University, Matsuyama 790-8566, Ehime, Japan

**Keywords:** *Cuminum cyminum* L., ovalbumin, allergic rhinitis, T-helper cells, T cell balance

## Abstract

*Cuminum cyminum* L. (cumin) seeds are widely used as a spice. Although we previously reported that the aqueous extract of cumin seeds suppresses the degranulation of rat basophilic RBL-2H3 cells, it has not been clarified whether the extract alleviates actual allergy symptoms in vivo. Therefore, in this study, we investigated the effect of oral administration of cumin seed aqueous extract (CAE) in ovalbumin (OVA)-induced allergic rhinitis. BALB/c mice were randomly divided into the following three groups: control group (five mice), OVA group (five mice), and OVA + CAE group (five mice). Allergic rhinitis was induced by sensitization (intraperitoneal, 25 μg OVA and 1.98 mg aluminum hydroxide gel) followed by challenge (intranasal, 400 μg OVA). The oral administration of CAE (25 mg/kg) reduced the sneezing frequency of OVA-induced allergic rhinitis model mice. In addition to reducing the serum immunoglobulin E and IL-4 levels, the oral administration of CAE reduced the production of T-helper type-2 (Th2) cytokines (IL-4, IL-5, IL-10, and IL-13) in the splenocytes of the model mice. Furthermore, a significant increase in the ratio of Th1 to Th2 cells was observed in the CAE-administered group. Our findings suggest that the ingestion of CAE improves T cell balance, the dominant state of Th2, and alleviates allergic rhinitis symptoms.

## 1. Introduction

Allergies are caused by an excessive reaction of the immune system to harmless substances (for example, pollen and food molecules) and are classified into four types (types I to IV) according to the mechanism of action. Type-I allergies, including hayfever, allergic rhinitis, and food allergies, involve a reaction called degranulation that occurs in mast cells and basophils. In response to cross-linking between the Fc receptor on the cell surface and immunoglobulin E (IgE)-antigen, the intracellular granules are released from the cells [1,2]. Chemical mediators, including histamine and leukotrienes, in granules cause immediate allergy symptoms by stimulating nerves and dilating blood vessels [3]. In addition, cytokines such as interleukin (IL)-33, IL-25 and thymic stromal lymphopoietin (TSLP) are produced from mucosal epithelial cells when stimulated by antigens; additionally, these cytokines act on activated T-helper type-2 (Th2) cells to secrete Th2-type cytokines such as IL-4, IL-5, and IL-13, thereby increasing the allergic response [4,5,6,7,8]. Recent studies have shown that activation of the phosphatidylinositol-3- kinase (PI3K), mitogen-activated protein kinase (MAPK), and nuclear transcription factor κB (NF-κB) pathways contributes to T cell activation, function, and differentiation, leading to eosinophilic recruitment and T-helper type-1 (Th1)/Th2 imbalance [9,10,11,12,13]. In addition, the treatment of inhibitors showed a reduction in Th2 cytokine production and the degree of allergic symptoms [11,12,13]. Therefore, the inhibition of Th2 cytokine secretion and improvement of T cell immune imbalance, as well as inhibition of histamine release and its action, could lead to preventive and therapeutic strategies for treating allergic diseases.

*Cuminum cyminum* L. (cumin) belongs to the *Apiaceae* (*Umbelliferae*) family; cumin seeds have a unique aroma and are widely used as a spice. The essential oil obtained from the seeds contains volatile components such as cuminaldehyde (4-isopropylbenzaldehyde), γ-terpinene, and β-pinene [14]. The essential oils and cuminaldehyde have been reported to have many biological functions, including anti-inflammatory, anti-diabetic, anti-oxidative, and anti-bacterial activities [14,15,16,17,18,19]. However, there are few reports concerning the functions of the water-soluble components in cumin seeds. We previously reported that an aqueous extract from cumin seed suppressed the antigen-induced degranulation of rat basophilic leukemia RBL-2H3 cells and passive cutaneous anaphylaxis (PCA) reaction in mice [20]. These findings suggest that the extract may alleviate allergic symptoms caused by histamine. In allergic reactions, not only chemical mediators released by degranulation but also allergy-related cytokines produced by immune cells, such as Th1 and Th2 cells, are involved [21,22]. However, it has not been clarified whether the aqueous extract from cumin seeds affects the secretion of cytokines involved in allergic reactions and actual allergy symptoms. Since cumin seed extract has been shown to downregulate PI3K phosphorylation [20], it may also exert inhibitory effects on T cell activation and differentiation by inhibiting the PI3K pathway. Hence, in this study, we investigated whether ingestion of cumin seed extract alleviates allergic symptoms and improves the secretion of T cell cytokines and T cell imbalance in an ovalbumin (OVA)-induced allergic rhinitis mouse.

## 2. Materials and Methods

### 2.1. Reagents

Albumin from chicken egg white (OVA), fetal bovine serum (FBS), penicillin, Roswell Park Memorial Institute 1640 (RPMI 1640) medium, and streptomycin were purchased from Sigma-Aldrich (St. Louis, MO, USA). Purified rat anti-mouse IgE, biotin rat anti-mouse IgE, goat affinity purified antibody to mouse IgG, and biotin-XX goat anti-mouse IgG_1_ were purchased from BD Biosciences (Franklin Lakes, NJ, USA), MP Biomedicals (Santa Ana, CA, USA), and Life Technologies (Carlsbad, CA, USA), respectively.

### 2.2. Sample Preparation

Cumin seed powder imported from Turkey was purchased by S&B Foods Inc, (Tokyo, Japan). A sodium phosphate buffer (NaPB; pH 7.4) was used as an extraction solvent to prevent the pH from changing significantly depending on the environment and sample concentration during sample storage or use. The powder was suspended in 10 mM NaPB at 0.05 g/mL, and the suspension was stirred at 30 rpm for 24 h using an RT-30mini (TAITEC Corporation, Saitama, Japan) in a low-temperature room at 12 °C. After centrifugation at 15,000× *g* and 4 °C for 20 min, the supernatant was collected, adjusted to a pH of 7.4, filtered using a 0.2 μm membrane for sterilization, and used as cumin seed aqueous extract (CAE).

### 2.3. Animals

BALB/c mice were acquired from Clea Japan (Tokyo, Japan) and housed in an animal room in the following conditions: a temperature of 24 ± 1 °C and 12 h alternating light and dark. All mice were fed with a standard diet and water ad libitum and acclimatized for 1 week before experiments. Animal experiments were approved by the Animal Experiment Committee of Ehime University and were performed in accordance with the Guidelines of Animal Experiments of Ehime University (approval number: 08U24-1).

### 2.4. Mouse Model of OVA-Induced Allergic Rhinitis

A mouse model of OVA-induced allergic rhinitis was established, as shown in Figure 1a. Six-week-old female BALB/c mice (15 mice) were randomly divided into the following three groups: control group (5 mice), OVA group (5 mice), and OVA + CAE group (5 mice). The mice of the OVA and OVA + CAE groups were sensitized with intraperitoneal injections of 200 μL phosphate buffered saline (PBS) containing 25 μg OVA and 1.98 mg aluminum hydroxide gel (Fujifilm Wako Pure Chemical, Osaka, Japan) on days 0, 7, and 14. After 1 week (day 21), the mice of the OVA and OVA + CAE groups were challenged with intranasal administration of 10 μL of PBS containing 400 μg OVA for 8 consecutive days from day 21 to 28. The mice of the control group were neither sensitized nor challenged with OVA but were injected intranasally with PBS only on day 28. The mice of the OVA + CAE group were orally administered CAE (25 mg/kg) for 8 consecutive days from day 21 to 28, while the mice of the control and OVA groups were orally administered 10 mM NaPB as a vehicle. The dosage of CAE was based on a previous study [20]. On day 28, the mice were counted for the frequency of sneezing and rubbing for 20 min after the challenge with OVA or PBS. On day 29, the mice were sacrificed, and blood and spleen were collected for further analysis.

### 2.5. Biochemical Measurement in the Serum of OVA-Induced Allergic Rhinitis Mice

On day 29, all mice were anesthetized with isoflurane, and blood was collected through cardiac puncture in a MiniCollect II tube (Greiner Bio-One, Kremsmünster, Austria). The tube was centrifuged at 3000× *g* and 15 °C for 10 min to obtain the serum. The samples were stored at −80 °C until further analysis. The concentrations of total IgE and total IgG_1_ were evaluated using an in-house-developed ELISA as described previously [23]. The concentrations of IL-4 and interferon gamma (IFN-γ) were evaluated using respective ELISA quantitation kits (eBioscience, San Diego, CA, USA) as per the manufacturer’s protocols. The absorbance values at 415 nm for total IgE and IgG_1_ and at 450 nm for the other studied factors were measured using an iMark microplate reader (Bio-Rad Laboratories, Hercules, CA, USA), and used to calculate the final concentrations.

### 2.6. Preparation of Splenocytes

After blood collection, the spleen was harvested and kept in sterile PBS until preparation. The spleen was passed through a 40 μm cell strainer to obtain a single cell suspension, and then the suspension underwent red blood cell lysis. The obtained splenocytes were seeded at 1.25×10^6^ cells/well to a 48-well culture plate and cultured in 5% FBS-RPMI 1640 containing 100 μg/mL OVA for 72 h at 37 °C. The concentrations of total IgE and total IgG_1_ in the cultured media were detected using an in-house-developed ELISA, as described previously [23]. IL-4, IL-5, IL-10, IL-13, and IFN-γ in culture media were detected by the respective ELISA quantitation kits as per the manufacturer’s protocols. The absorbance values at 415 nm for total IgE and IgG_1_ and at 450 nm for the other studied factors were measured using an iMark microplate reader and were used to calculate the final concentrations.

### 2.7. Flow Cytometric Analysis

The effect of CAE on T cell balance in the spleen of OVA-induced allergic rhinitis mice was examined in another experiment in order to obtain a greater difference. The OVA-induced model mice were established as described above, and CAE was orally administered from days 0 to 27 (Figure 1b). On day 28, the splenocytes were prepared as described above and stained using the Mouse Th1/Th2/Th17 Phenotyping Kit (BD Biosciences, Franklin Lakes, NJ, USA). Briefly, the splenocytes were seeded at 2.0 × 10^6^ cells/well to a 24-well culture plate and cultured in 5% FBS-RPMI 1640 containing 50 ng/mL phorbol 12-myristate 13-acetate (PMA) and 1 μg/mL ionomycin for 5 h at 37 °C. After incubation, the cells were fixed, permeabilized, and stained according to the protocol using each reagent provided in the kit. The stained cells were suspended with 2% FBS-PBS prior to flow cytometric analysis. To detect Th1 and Th2 cells, the splenocytes were stained with fluorescein isothiocyanate (FITC)-labeled anti-IFN-γ, allophycocyanin (APC)-labeled anti-IL-4, and peridinin chlorophyll protein-cyanine5.5 (PerCP-Cy5.5)-labeled anti-CD4. The distribution of Th1 cells (IFN-γ^+^/CD4^+^) and Th2 cells (IL-4^+^/CD4^+^) was analyzed using a FACSVerse flow cytometer (Becton Dickinson, Franklin Lakes, NJ, USA). The CD4^+^ cell population was gated among 10,000 cells, and the proportion of IFN-γ^+^ or IL-4^+^ cells was detected among them.

### 2.8. Statistical Analysis

Data are presented as the mean ± standard error of the mean (SEM). Statistical analysis was performed using GraphPad Prism version 8.4.3 (GraphPad Software, San Diego, CA, USA). Data were analyzed using one-way analysis of variance followed by post-hoc Dunnett’s test. A value of *p* < 0.05 was considered statistically significant.

## 3. Results

### 3.1. Effect of Oral Administration of CAE on Allergy Symptoms in OVA-Induced Allergic Rhinitis Mice

Body weight loss due to allergy induction was observed in the OVA and OVA + CAE groups, whereas there was no significant difference between the groups (Table 1). There was a significant difference in the sneezing frequency between the control and OVA groups (*p* < 0.01), and the OVA + CAE group showed a decreasing tendency (*p* = 0.076) compared to the OVA group. In contrast, the rubbing frequency increased in the OVA group, but the differences between the OVA and OVA + CAE groups were not significant (*p* = 0.834).

### 3.2. Effect of CAE on Serum Levels of Immunoglobulins and Cytokines of OVA-Induced Allergic Rhinitis Mice

Serum IgE, IgG_1_, and IL-4 levels in the OVA group were significantly increased by OVA injection (Table 2). In contrast, the administration of CAE significantly reduced the serum levels of IgE (*p* < 0.01) and IL-4 (*p* < 0.05). There was no difference in the serum level of IgG_1_ between the OVA and OVA + CAE groups. Although the serum level of IFN-γ was not significantly different between these groups, it increased in the OVA + CAE group compared to the OVA group (*p* = 0.325).

### 3.3. Effect of CAE on Production of Immunoglobulins and Cytokines in Splenocytes of OVA-Induced Allergic Rhinitis Mice

The production of IgE and IgG_1_ in the OVA group significantly increased following OVA injection (Table 3). In the OVA + CAE group, the IgG_1_ production was not significantly affected. In addition, there was a slight decrease in IgE production; however, no significant difference (*p* = 0.207) was detected from the OVA group. IL-10 production was high in the OVA group, whereas it was significantly reduced in the OVA + CAE group (*p* < 0.05). IFN-γ production increased in the OVA + CAE group compared to that in the OVA group, and it was similar to that in the control group, though not significantly (*p* = 0.650). Conversely, the production of IL-4 (*p* < 0.05), IL-5 (*p* < 0.01), and IL-13 (*p* = 0.066) reduced in the OVA + CAE group.

### 3.4. Effect of CAE on Th1/Th2 Balance

The proportion of Th1/Th2 cells in the spleen was analyzed by performing flow cytometry. In this experiment, the oral administration period of CAE was changed from that used at the start of the experiment to obtain a greater difference (Figure 1b). The proportion of Th1 cells (IFN-γ^+^/CD4^+^ cells) was slightly increased in the OVA + CAE group compared to that in the OVA group, but there was no significant difference (*p* = 0.1767) (Figure 2). On the other hand, the proportion of Th2 cells (IL-4^+^/CD4^+^ cells) did not decrease significantly (*p* = 0.4867). A significant increase in the Th1/Th2 ratio was confirmed in the OVA + CAE group compared to the OVA group (*p* < 0.05).

## 4. Discussion

The in vivo experiments showed that the oral administration of CAE reduced the sneezing frequency in mice with allergic rhinitis induced by OVA (Table 1). In addition, the serum IgE, serum IL-4, and Th2 cytokines (IL-4, IL-5, IL-10, and IL-13) produced by splenocytes were significantly reduced in the OVA + CAE group (Table 2 and Table 3). IL-4 and IL-13 are anti-inflammatory cytokines produced by activated Th2 cells. These cytokines are associated with the differentiation and maturation of B cells, which promote the expression of CD23 and MHC class II molecules and class switching to IgE [24]. IL-5 is a hematopoietic cytokine that plays an important role in the differentiation, maturation, recruitment, and activation of eosinophils at allergic inflammation sites [25]. The suppression of Th2 cytokine production from Th2 cells by the oral administration of CAE likely regulates the class switch to IgE and the infiltration of immune cells into inflammation sites, leading to a decrease in serum IgE levels and the reduction in allergy symptoms. IL-10 production in splenocytes was also reduced in the OVA + CAE group (Table 3). IL-10 is an anti-inflammatory cytokine produced by Th2 cells that regulates abnormal immune responses by acting on the activation and antigen-presenting ability of monocytes and T cells exposed to antigens. However, studies have also reported that IL-10 is essential for inducing allergy symptoms [26,27]. In our study, OVA-induced allergic rhinitis symptoms were alleviated. Therefore, it is unlikely that the suppression of IL-10 production had a negative effect. Despite the many studies conducted on this topic, the action and regulation of IL-10 in allergic reactions has not yet been elucidated; however, our results lead us to speculate that the suppression of IL-10 production by the oral administration of CAE may be involved in the alleviation of OVA-induced allergy symptoms.

The level of IFN-γ in both serum and splenocytes was slightly increased in the OVA + CAE group, suggesting that the oral administration of CAE may regulate the dominant state of Th2 cells. A significant increase in the Th1/Th2 ratio was observed in the OVA + CAE group (Figure 2). IFN-γ and IL-12 play important roles in the differentiation of Th1 cells; IL-12 is a heterodimeric pro-inflammatory cytokine that induces IFN-γ production and Th1 cell differentiation [28]. The differentiation of Th1 cells by IL-12 occurs via the activation of the transcription factor STAT4, which induces IFN-γ production [29]. The increase in the IFN-γ production in the splenocytes appeared to be slightly correlated with an increased proportion of Th1 cells in the spleen. In contrast, the production of Th2 cytokines in the splenocytes decreased, whereas the proportion of Th2 cells did not decrease. These results suggest that CAE suppresses the cytokine secretion ability of Th2 cells but does not affect the proportion of Th2 cells in the spleen; furthermore, it improves the Th1/Th2 balance by increasing the proportion of Th1 cells, resulting in the alleviation of allergic symptoms. PI3K/AKT, MAPK, and NF-κB pathways are known to be involved in T cell activation and differentiation, contributing to allergic inflammation by promoting Th1/Th2 imbalance [9,10,11,12,13]. Our previous study suggested that cumin seed aqueous extract contains substances downregulating PI3K activation [20]. Thus, the active substances in CAE may suppress T cell activation and differentiation by directly or indirectly inhibiting the PI3K pathway. Further research is needed to identify the active substances in CAE regulating T cell balance and to elucidate their mechanism of action.

The anti-allergic effect of CAE was also exerted in vivo, suggesting that the active substance is efficiently digested and absorbed when ingested orally. In our previous study, we identified the active substances contained in cumin seeds that inhibit degranulation in RBL-2H3 cells, i.e., apigenin, luteolin, and umbelliferose [30]. Apigenin and luteolin have also been reported to attenuate OVA-induced allergic rhinitis [31,32,33,34]. However, since apigenin and luteolin are practically insoluble in water, they are hardly extracted in CAE. Umbelliferose, a characteristic trisaccharide present in plants of the *Apiaceae* (*Umbelliferae*) family, has been generally recognized as an indigestible oligosaccharide. However, some studies have demonstrated that dietary raffinose, which is an isomer of umbelliferose, is absorbed intact in the gastrointestinal tract and exerts an anti-allergic effect [35,36]. Therefore, it is possible that umbelliferose is also absorbed from the gastrointestinal tract and exerts an anti-allergic effect in vivo. On the other hand, CAE dialyzed with a membrane of a molecular weight cut off of 500 showed a partial decrease in anti-degranulation activity compared to non-dialyzed CAE; however, the activity was still detected (Appendix A). This result suggests that CAE contains not only active substances with a molecular weight of 500 or less, but also active substances of a high molecular weight. In addition, the active substances identified in cumin seeds mentioned above are present only in small quantities, suggesting that interactions between water-soluble substances, including substances that have yet to be identified, are possibly responsible for the observed strong activities of CAE. The water-soluble cumin seed ingredients need to be further analyzed to obtain further insight into their anti-allergic properties.

In conclusion, our findings suggest that the ingestion of CAE improves the T-cell balance and alleviates allergy symptoms of allergic rhinitis in vivo. This study, which focused on the water-soluble ingredients in cumin seeds that have not previously received much attention, can contribute to the understanding of the value of cumin seeds and to the library of food-derived functional ingredients, thereby developing functional food science. In addition, cumin seeds are widely used as a spice, and the volatile non-polar ingredients in the seeds are extracted as essential oils for a range of uses. However, many promising water-soluble ingredients that remain in the seed are wasted after extraction. We believe that these findings provide insights into the biological functions of cumin seed ingredients in the field of functional foods and contribute to the effective utilization of cumin seed residue after oil extraction.

## Figures and Tables

**Figure 1 foods-11-03224-f001:**
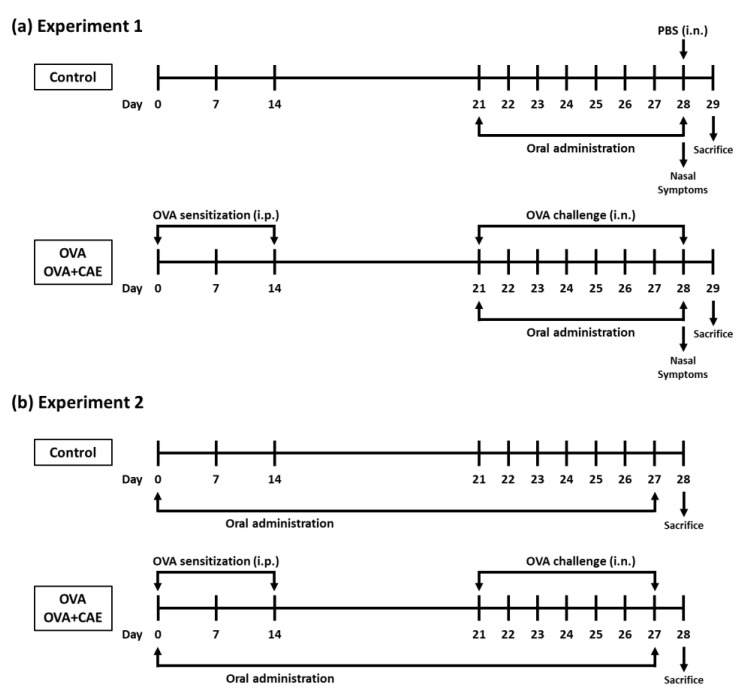
Experimental design used for inducing allergic rhinitis in mice. Abbreviations used: OVA: ovalbumin; PBS: phosphate buffered saline; CAE: cumin seed aqueous extract; i.p.: intraperitoneally; i.n.: intranasally. (**a**) Experiment 1 was conducted to examine the effect of CAE on OVA challenge-induced allergic rhinitis symptoms and biological parameters, and (**b**) Experiment 2 to examine the effect of CAE on T cell balance in OVA-induced allergic rhinitis mice.

**Figure 2 foods-11-03224-f002:**
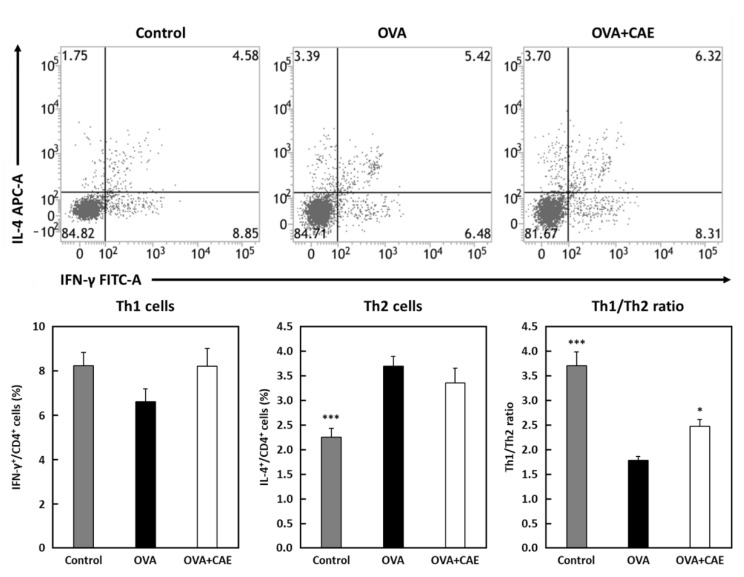
The percentage of Th1 and Th2 cells and Th1/Th2 ratio in spleen. Abbreviations used: OVA: ovalbumin; CAE: cumin seed aqueous extract; SEM: standard error of the mean. Splenocytes were cultured with phorbol 12-myristate 13-acetate (PMA) and ionomycin for 5 h at 37 °C. The distribution of Th1 cells (IFN-γ^+^/CD4^+^) and Th2 cells (IL-4^+^/CD4^+^) were analyzed using a flow cytometer. The Th1/Th2 ratio was observed by the ratio of the percentage of Th1 cells to the percentage of Th2 cells. Gray bar, closed bar, and open bar represent control, OVA, and OVA + CAE groups, respectively. Data are expressed as means ± SEM (*n* = 7). * *p* < 0.05 and *** *p* < 0.001 against OVA group by post hoc Dunnett’s test.

**Table 1 foods-11-03224-t001:** Effect of CAE administration on body weight and OVA challenge-induced nasal sneezing and rubbing in allergic rhinitis mice.

Parameters	Control	OVA	OVA + CAE	OVA vs. Control *p*-Value	OVA vs. OVA + CAE *p*-Value
	Mean ± SEM (*n* = 5)				
Body weight (g) on day 29	23.6 ± 0.3	21.1 ± 0.7	21.0 ± 0.5	0.010	0.995
OVA challenge on day 28					
Sneezing (number)	6.8 ± 1.6	50.0 ± 10.8	27.0 ± 6.0	0.002	0.076
Rubbing (number)	21.6 ± 7.9	44.2 ± 5.2	38.8 ± 9.0	0.098	0.834

Abbreviations used: OVA, ovalbumin; CAE, cumin seed aqueous extract; SEM, standard error of the mean. Data were analyzed by one-way analysis of variance followed by post-hoc Dunnett’s test.

**Table 2 foods-11-03224-t002:** Effect of CAE administration on OVA-induced alterations in serum levels of immunoglobulins and cytokines in allergic rhinitis mice.

Serum Levels	Control	OVA	OVA + CAE	OVA vs. Control *p*-Value	OVA vs. OVA + CAE *p*-Value
	Mean ± SEM (*n* = 5)				
IgE (μg/mL)	0.9 ± 0.3	41.9 ± 3.7	29.3 ± 1.4	<0.001	0.004
IgG_1_ (mg/mL)	0.8 ± 0.2	5.9 ± 0.2	5.6 ± 0.3	<0.001	0.438
IL-4 (pg/mL)	1.3 ± 0.2	9.5 ± 2.4	3.7 ± 0.6	0.003	0.023
IFN-γ (pg/mL)	769.5 ± 344.9	62.0 ± 37.3	716.1 ± 474.2	0.276	0.325

Abbreviations used: OVA, ovalbumin; CAE, cumin seed aqueous extract; SEM, standard error of the mean; IgE, immunoglobulin E; IgG_1_, immunoglobulin G_1_; IL-4, interleukin-4; IFN-γ, interferon-γ. Data were analyzed by one-way analysis of variance followed by post-hoc Dunnett’s test.

**Table 3 foods-11-03224-t003:** Effect of CAE administration on OVA-induced alterations in the production of immunoglobulins and cytokines in splenocytes of allergic rhinitis mice.

Serum Levels	Control	OVA	OVA + CAE	OVA vs. Control *p*-Value	OVA vs. OVA + CAE *p*-Value
	Mean ± SEM (*n* = 5)				
IgE (ng/mL)	6.5 ± 2.1	37.7 ± 6.7	24.4 ± 6.9	0.004	0.207
IgG_1_ (ng/mL)	7.9 ± 4.5	131.1 ± 5.0	124.9 ± 10.3	<0.001	0.767
IL-4 (pg/mL)	7.6 ± 0.6	875.6 ± 178.3	381.2 ± 121.1	<0.001	0.029
IL-5 (pg/mL)	8.7 ± 0.3	744.9 ± 73.7	331.0 ± 96.0	<0.001	0.002
IL-10 (pg/mL)	31.7 ± 4.0	829.9 ± 47.8	556.3 ± 118.9	<0.001	0.041
IL-13 (ng/mL)	0.1 ± 0.0	3.3 ± 0.4	2.0 ± 0.6	<0.001	0.066
IFN-γ (ng/mL)	1.7 ± 0.4	1.0 ± 0.2	1.5 ± 0.6	0.429	0.650

Abbreviations used: OVA: ovalbumin; CAE: cumin seed aqueous extract; SEM: standard error of the mean; IgE: immunoglobulin E; IgG_1_: immunoglobulin G_1_; IL-4: interleukin-4; IL-5: interleukin-5; IL-10: interleukin-10; IL-13: interleukin-13; IFN-γ: interferon-γ. Data were analyzed using one-way analysis of variance followed by post-hoc Dunnett’s test.

## Data Availability

The data that support the findings in this study are available from the corresponding author upon reasonable request.

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
