# Peer review of "Aqueous Extract from *Cuminum cyminum* L. Seed Alleviates Ovalbumin-Induced Allergic Rhinitis in Mouse via Balancing of Helper T Cells"

_foods, 2022, doi:10.3390/foods11203224_

Round 1

Reviewer 1 Report

Some observations are highlighted in the Methodology

item 2.2 - Please describe the origin of the cumin seed, providing information as to whether it was purchased at the local market or collected and dried. Indicate which city and country. To prepare the aqueous extract, the suspension was incubated at 12 °C for 24 hours, but in which equipment was this incubation performed? How was the homogenization of the solution performed?

item 2.4 - The explanation of the experimental design is incomprehensible. It is unclear how many animals there are in total and what treatment each group received. Of the 15 animals in the experiment, 5 were sensitized, challenged and treated; 5 were sensitized but not challenged or treated and received oral NaPB. Another 5 animals were not sensitized, challenged or treated and received oral NaPB. It would be more appropriate to have a fourth group of animals sensitized, challenged but not treated. know what level or type of response the animals would develop to the challenge, but without having been treated with CAE.

Please what is the definition of NaPB?

Figure 1 - Like the text, the figure is not clear.

Please transcribe the description of the splenocyte culture and the interferon gamma quantification method in a separate item.

item 2.5 - The sentence that explains that the animals were sensitized and challenged, has already been described in the previous item, it is not necessary to repeat. Describe how the flow cytometry technique was performed in more detail.

Discussion

"On the other hand, CAE dialyzed with a membrane of molecular weight cut off 500 showed a partial decrease in anti-degranulation activity compared to non-dialyzed CAE, but the activity remained (data not shown). This result suggests that CAE contains not only active substances with molecular weight of 500 or less, but also high molecular active substances." I consider this information relevant for the explanation of your results. Therefore, I suggest describing this step in the methodology and in the result.

Reviewer 2 Report

Regarding the manuscript entitled ‘’ Aqueous extract from Cuminum cyminum L. seed alleviates ovalbumin-induced allergic rhinitis in mouse via balancing of helper T cells’’

Abstract.

More details about the design and number of animals should be added.

Introduction

Please add a hypothesis.

Materials and Methods

The chemical composition of the extract should be added, what are the active compounds?

L78. Of the experiment?

L79. A small set of animals!!

L81. Days of what? The time point should be clarified clearly it is a little bit confusing.

L83. On which basis the authors chose this dose?

For Results

Please add p-value for significant findings.

Figures. why these data are not presented as charts or even add data in tables to be clear for the reader? The data should present the means and SE or SEM.

How did the authors confirm these findings with the small number of animals?

L165. Tendency, no information about the trends in statistical analysis section.

Discussion

Mechanism of action, how the active compounds in the cumin extract affect the studied inflammatory cytokines?

Reviewer 3 Report

The study is well design to prove the anti allergic rhinitis effects of aqueous extract from cuminum cyminum L. seed. the author has a few comments. 

1.Figure 1 is composed of a, and b. it should be described in the method part and the explanation should be clear. Also, Figure should be understandable as itself so the figure 1 need some footnote to present details. 

2. The author selected water extract and some flavonoids have been found in CAE showing anti-allergic rhinitis. Does author have specific rational to select water extract to conduct research? For example, water extract showed  greater effect than ethanol extracts. The reviewer has concern ethanol extract may shows better effects and bioactive compounds in water extracts are not much different which means mainly not but some extracted by water.  Author should thought about this and mention the rational of the extraction method in the manuscript.

Round 2

Reviewer 2 Report

Thank you for the revisions